# GLoMo: Unsupervised Learning of Transferable Relational Graphs

**Zhilin Yang**[*1], **Jake (Junbo) Zhao**[*23], **Bhuwan Dhingra**[1]
**Kaiming He**[3], **William W. Cohen**[1], **Ruslan Salakhutdinov**[1], **Yann LeCun**[23]
[*]Equal contribution
[1]Carnegie Mellon University, [2]New York University, [3]Facebook AI Research
{zhiliny,bdhingra,wcohen,rsalakhu}@cs.cmu.edu
{jakezhao,yann}@cs.nyu.com, kaiminghe@fb.com

## Abstract

Modern deep transfer learning approaches have mainly focused on learning *generic* feature vectors from one task that are transferable to other tasks, such as word embeddings in language and pretrained convolutional features in vision. However, these approaches usually transfer unary features and largely ignore more structured graphical representations. This work explores the possibility of learning generic *latent relational graphs* that capture dependencies *between* pairs of data units (e.g., words or pixels) from large-scale unlabeled data and transferring the graphs to downstream tasks. Our proposed transfer learning framework improves performance on various tasks including question answering, natural language inference, sentiment analysis, and image classification. We also show that the learned graphs are generic enough to be transferred to different embeddings on which the graphs have not been trained (including GloVe embeddings, ELMo embeddings, and task-specific RNN hidden units), or embedding-free units such as image pixels.

## 1 Introduction

Recent advances in deep learning have largely relied on building blocks such as convolutional networks (CNNs) [19] and recurrent networks (RNNs) [14] augmented with attention mechanisms [1]. While possessing high representational capacity, these architectures primarily operate on grid-like or sequential structures due to their built-in "innate priors". As a result, CNNs and RNNs largely rely on high expressiveness to model complex structural phenomena, compensating the fact that they do not explicitly leverage structural, graphical representations.

This paradigm has led to a standardized norm in transfer learning and pretraining—fitting an expressive function on a large dataset with or without supervision, and then applying the function to downstream task data for feature extraction. Notable examples include pretrained ImageNet features [13] and pretrained word embeddings [24, 29].

In contrast, a variety of real-world data exhibit much richer relational *graph* structures than the simple grid-like or sequential structures. This is also emphasized by a parallel work [3]. For example in the language domain, linguists use parse trees to represent syntactic dependency between words; information retrieval systems exploit knowledge graphs to reflect entity relations; and coreference resolution is devised to connect different expressions of the same entity. As such, these exemplified structures are universally present in almost any natural language data regardless of the target tasks, which suggests the possibility of transfer across tasks. These observations also generalize to other domains such as vision, where modeling the relations between pixels is proven useful [28, 50, 44]. One obstacle remaining, however, is that many of the universal structures are essentially human-

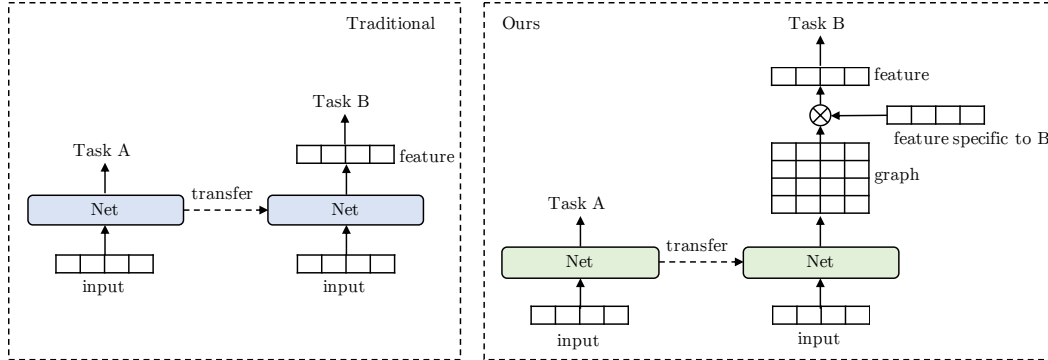

Figure 1: Traditional transfer learning versus our new transfer learning framework. Instead of transferring features, we transfer the graphs output by a network. The graphs are multiplied by task-specific features (e.g. embeddings or hidden states) to produce structure-aware features.

curated and expensive to acquire on a large scale, while automatically-induced structures are mostly limited to one task [17, 42, 44].

In this paper, we attempt to address two challenges: 1) to break away from the standardized norm of feature-based deep transfer learning[1], and 2) to learn versatile structures in the data with a data-driven approach. In particular, we are interested in learning transferable *latent relational graphs*, where the nodes of a latent graph are the input *units*, e.g., all the words in a sentence. The task of latent relational graph learning is to learn an *affinity matrix* where the weights (possibly zero) capture the dependencies between any pair of input units.

To achieve the above goals, we propose a novel framework of unsupervised latent graph learning called GLoMo (Graphs from LOw-level unit MOdeling). Specifically, we train a neural network from large-scale unlabeled data to output latent graphs, and transfer the network to extract graph structures on downstream tasks to augment their training. This approach allows us to separate the features that represent the semantic meaning of each unit and the graphs that reflect how the units may interact. Ideally, the graphs capture task-independent structures underlying the data, and thus become applicable to different sets of features. Figure 1 highlights the difference between traditional feature-based transfer learning and our new framework.

Experimental results show that GLoMo improves performance on various language tasks including question answering, natural language inference, and sentiment analysis. We also demonstrate that the learned graphs are generic enough to work with various sets of features on which the graphs have not been trained, including GloVe embeddings [29], ELMo embeddings [30], and task-specific RNN states. We also identify key factors of learning successful generic graphs: decoupling graphs and features, hierarchical graph representations, sparsity, unit-level objectives, and sequence prediction. To demonstrate the generality of our framework, we further show improved results on image classification by applying GLoMo to model the relational dependencies between the pixels.

## 2 Unsupervised Relational Graph Learning

We propose a framework for unsupervised latent graph learning. Given a one-dimensional input $x = (x_1, \cdots, x_T)$, where each $x_t$ denotes an input unit at position $t$ and $T$ is the length of the sequence, the goal of latent graph learning is to learn a $(T \times T)$ affinity matrix $\mathbf{G}$ such that each entry $G_{ij}$ captures the dependency between the unit $x_i$ and the unit $x_j$. The affinity matrix is asymmetric, representing a directed weighted graph. In particular, in this work we consider the case where each column of the affinity matrix sums to one, for computational convenience. In the following text, with a little abuse of notation, we use $\mathbf{G}$ to denote a set of affinity matrices. We use the terms "affinity matrices" and "graphs" interchangeably.

During the unsupervised learning phase, our framework trains two networks, a *graph predictor* network $g$ and a *feature predictor* network $f$. Given the input $x$, our graph predictor $g$ produces a set of graphs $\mathbf{G} = g(x)$. The graphs $\mathbf{G}$ are represented as a 3-d tensor in $\mathbb{R}^{L \times T \times T}$, where $L$ is the

number of *layers* that produce graphs. For each layer $l$, the last two dimensions $\mathbf{G}^l$ define a $(T \times T)$ affinity matrix that captures the dependencies between any pair of input units. The feature predictor network $f$ then takes the graphs $\mathbf{G}$ and the original input $x$ to perform a predictive task.

During the transfer phase, given an input $x'$ from a downstream task, we use the graph predictor $g$ to extract graphs $\mathbf{G} = g(x')$. The extracted graphs $\mathbf{G}$ are then fed as the input to the downstream task network to augment training. Specifically, we multiply $\mathbf{G}$ with task-specific features such as input embeddings and hidden states (see Figure 1). The network $f$ is discarded during the transfer phase.

Next, we will introduce the network architectures and objective functions for unsupervised learning, followed by the transfer procedure. An overview of our framework is illustrated in Figure 2.

## 2.1 Unsupervised Learning

**Graph Predictor** The graph predictor $g$ is instantiated as two multi-layer CNNs, a *key* CNN, and a *query* CNN. Given the input $x$, the key CNN outputs a sequence of convolutional features $(\mathbf{k}_1, \cdots, \mathbf{k}_T)$ and the query CNN similarly outputs $(\mathbf{q}_1, \cdots, \mathbf{q}_T)$. At layer $l$, based on these convolutional features, we compute the graphs as

$$G_{ij}^l = \frac{\left(\text{ReLU}(\mathbf{k}_i^{l\top}\mathbf{q}_j^l + b)\right)^2}{\sum_{i'}\left(\text{ReLU}(\mathbf{k}_{i'}^{l\top}\mathbf{q}_j^l + b)\right)^2} \tag{1}$$

where $\mathbf{k}_i^l = \mathbf{W}_k^l\mathbf{k}_i$ and $\mathbf{q}_j^l = \mathbf{W}_q^l\mathbf{q}_j$. The matrices $\mathbf{W}_k^l$ and $\mathbf{W}_q^l$ are model parameters at layer $l$, and the bias $b$ is a scalar parameter. This resembles computing the attention weights [1] from position $j$ to position $i$ except that the exponential activation in the softmax function is replaced with a squared ReLU operation—we use ReLUs to enforce sparsity and the square operations to stabilize training. Moreover, we employ convolutional networks to let the graphs $\mathbf{G}$ be aware of the local order of the input and context, up to the size of each unit's receptive field.

**Feature Predictor** Now we introduce the feature predictor $f$. At each layer $l$, the input to the feature predictor $f$ is a sequence of features $\mathbf{F}^{l-1} = (\mathbf{f}_1^{l-1}, \cdots, \mathbf{f}_t^{l-1})$ and an affinity matrix $\mathbf{G}^l$ extracted by the graph predictor $g$. The zero-th layer features $\mathbf{F}_0$ are initialized to be the embeddings of $x$. The affinity matrix $\mathbf{G}^l$ is then combined with the current features to compute the next-layer features at each position $t$,

$$\mathbf{f}_t^l = v(\sum_j G_{jt}^l\mathbf{f}_j^{l-1}, \mathbf{f}_t^{l-1}) \tag{2}$$

where $v$ is a compositional function such as a GRU cell [8] or a linear layer with residual connections. In other words, the feature at each position is computed as a weighted sum of other features, where the weights are determined by the graph $\mathbf{G}^l$, followed by transformation function $v$.

**Objective Function** At the top layer, we obtain the features $\mathbf{F}^L$. At each position $t$, we use the feature $\mathbf{f}_t^L$ to initialize the hidden states of an RNN decoder, and employ the decoder to predict the units following $x_t$. Specifically, the RNN decoder maximizes the conditional log probability $\log P(x_{t+1}, \cdots, x_{t+D}|x_t, \mathbf{f}_t^l)$ using an auto-regressive factorization as in standard language modeling [47] (also see Figure 2). Here $D$ is a hyper-parameter called the *context length*. The overall objective is written as the sum of the objectives at all positions $t$,

$$\max \sum_t \log P(x_{t+1}, \cdots, x_{t+D}|x_t, \mathbf{f}_t^L) \tag{3}$$

Because our objective is context prediction, we mask the convolutional filters and the graph $\mathbf{G}$ (see Eq. 1) in the network $g$ to prevent the network from accessing the future, following [34].

### 2.1.1 Desiderata

There are several key desiderata of the above unsupervised learning framework, which also highlight the essential differences between our framework and previous work on self-attention and predictive unsupervised learning:

**Decoupling graphs and features** Unlike self-attention [42] that fuses the computation of graphs and features into one network, we employ separate networks $g$ and $f$ for learning graphs and features

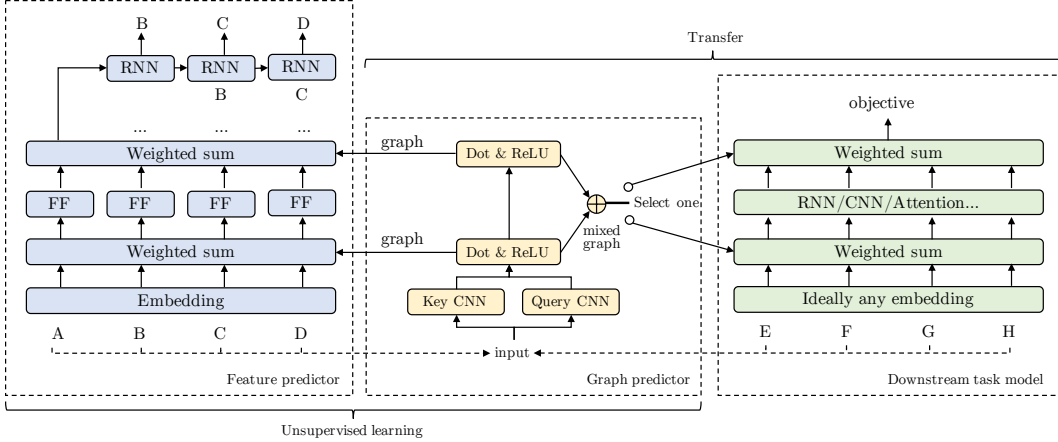

Figure 2: Overview of our approach GLoMo. During the unsupervised learning phase, the feature predictor and the graph predictor are jointly trained to perform context prediction. During the transfer phase, the graph predictor is frozen and used to extract graphs for the downstream tasks. An RNN decoder is applied to all positions in the feature predictor, but we only show the one at position "A" for simplicity. "Select one" means the graphs can be transferred to any layer in the downstream task model. "FF" refers to feed-forward networks. The graphs output by the graph predictor are used as the weights in the "weighted sum" operation (see Eq. 2).

respectively. The features represent the semantic meaning of each unit while the graph reflects how the units may interact. This increases the transferability of the graphs $\mathbf{G}$ because (1) the graph predictor $g$ is freed from encoding task-specific non-structural information, and (2) the decoupled setting is closer to our transfer setting, where the graphs and features are also separated.

**Sparsity** Instead of using Softmax for attention [1], we employ a squared ReLU activation in Eq. (1) to enforce sparse connections in the graphs. In fact, most of the linguistically meaningful structures are sparse, such as parse trees and coreference links. We believe sparse structures reduce noise and are more transferable.

**Hierarchical graph representations** We learn multiple layers of graphs, which allows us to model hierarchical structures in the data.

**Unit-level objectives** In Eq. (3), we impose a context prediction objective on each unit $x_t$. An alternative is to employ a sequence-level objective such as predicting the next sentence [18] or translating the input into another language [42]. However, since the weighted sum operation in Eq. (2) is permutation invariant, the features in each layer can be randomly shuffled without affecting the objective, which we observed in our preliminary experiments. As a result, the induced graph bears no relation to the structures underlying the input $x$ when a sequence-level objective is employed.

**Sequence prediction** As opposed to predicting just the immediate next unit [28, 30], we predict the context of length up to $D$. This gives stronger training signals to the unsupervised learner.

Later in the experimental section, we will demonstrate that all these factors contribute to successful training of our framework.

## 2.2 Latent Graph Transfer

In this section, we discuss how to transfer the graph predictor $g$ to downstream tasks.

Suppose for a downstream task the model is a deep multi-layer network. Specifically, each layer is denoted as a function $h$ that takes in features $\mathbf{H} = (\mathbf{h}_1, \cdots, \mathbf{h}_T)$ and possibly additional inputs, and outputs features $(\mathbf{h}'_1, \cdots, \mathbf{h}'_T)$. The function $h$ can be instantiated as any neural network component, such as CNNs, RNNs, attention, and feed-forward networks. This setting is general and subsumes the majority of modern neural architectures.

Given an input example $x'$ from the downstream task, we apply the graph predictor to obtain the graphs $\mathbf{G} = g(x')$. Let $\mathbf{\Lambda}^l = \prod_{i=1}^{l} \mathbf{G}^i \in \mathbb{R}^{T \times T}$ denote the product of all affinity matrices from the first layers to the $l$-th layer. This can be viewed as propagating the connections among multiple layers of graphs, which allows us to model hierarchical structures. We then take a mixture of all the graphs

Table 1: Main results on natural language datasets. Self-attention modules are included in all baseline models. All baseline methods are feature-based transfer learning methods, including ELMo and GloVe. Our methods combine graph-based transfer with feature-based transfer. Our graphs operate on various sets of features, including GloVe embeddings, ELMo embeddings, and RNN states. "mism." refers to the "mismatched" setting.

| Transfer method | SQuAD GloVe | | SQuAD ELMo | | IMDB GloVe | MNLI GloVe | |
| | EM | F1 | EM | F1 | Accuracy | matched | mism. |
| --- | --- | --- | --- | --- | --- | --- | --- |
| transfer feature only (baseline) | 69.33 | 78.73 | 74.75 | 82.95 | 88.51 | 77.14 | 77.40 |
| GLoMo on embeddings | 70.84 | 79.90 | **76.00** | **84.13** | **89.16** | **78.32** | **78.00** |
| GLoMo on RNN states | **71.30** | **80.24** | 76.20 | 83.99 | - | - | - |

in $\{\mathbf{G}^l\}_{l=1}^L \cup \{\mathbf{\Lambda}^l\}_{l=1}^L$,

$$\mathbf{M} = \sum_{l=1}^L m_G^l \mathbf{G}^l + \sum_{l=1}^L m_\Lambda^l \mathbf{\Lambda}^l, \quad \text{s.t.} \quad \sum_{l=1}^L (m_G^l + m_\Lambda^l) = 1$$

The mixture weights $m_G^l$ and $m_\Lambda^l$ can be instantiated as Softmax-normalized parameters [30] or can be conditioned on the features $\mathbf{H}$. To transfer the mixed latent graph, we again adopt the weighted sum operation as in Eq. (2). Specifically, we use the weighted sum $\mathbf{HM}$ (see Figures 1 and 2), in addition to $\mathbf{H}$, as the input to the function $h$. This can be viewed as performing attention with weights given by the mixed latent graph $\mathbf{M}$. This setup of latent graph transfer is general and easy to be plugged in, as the graphs can be applied to any layer in the network architecture, with either learned or pretrained features $\mathbf{H}$, at variable length.

## 2.3 Extensions and Implementation

So far we have introduced a general framework of unsupervised latent graph learning. This framework can be extended and implemented in various ways.

In our implementation, at position $t$, in addition to predicting the forward context $(x_{t+1}, \cdots, x_{t+D})$, we also use a separate network to predict the backward context $(x_{t-D}, \cdots, x_{t-1})$, similar to [30]. This allows the graphs $\mathbf{G}$ to capture both forward and backward dependencies, as graphs learned from one direction are masked on future context. Accordingly, during transfer, we mix the graphs from two directions separately.

In the transfer phase, there are different ways of effectively fusing $\mathbf{H}$ and $\mathbf{HM}$. In practice, we feed the concatenation of $\mathbf{H}$ and a gated output, $\mathbf{W}_1[\mathbf{H}; \mathbf{HM}] \odot \sigma(\mathbf{W}_2[\mathbf{H}; \mathbf{HM}])$, to the function $h$. Here $\mathbf{W}_1$ and $\mathbf{W}_2$ are parameter matrices, $\sigma$ denotes the sigmoid function, and $\odot$ denotes element-wise multiplication. We also adopt the multi-head attention [42] to produce multiple graphs per layer. We use a mixture of the graphs from different heads for transfer.

It is also possible to extend our framework to 2-d or 3-d data such as images and videos. The adaptations needed are to adopt high-dimensional attention [44, 28], and to predict a high-dimensional context (e.g., predicting a grid of future pixels). As an example, in our experiments, we use these adaptations on the task of image classification.

## 3 Experiments

### 3.1 Natural Language Tasks and Setting

**Question Answering** The stanford question answering dataset [31](SQuAD) was recently proposed to advance machine reading comprehension. The dataset consists of more than 100,000+ question-answer pairs from 500+ Wikipedia articles. Each question is associated with a corresponding reading passage in which the answer to the question can be deduced.

**Natural Language Inference** We chose to use the latest Multi-Genre NLI corpus (MNLI) [46]. This dataset has collected 433k sentence pairs annotated with textual entailment information. It uses the same modeling protocol as SNLI dataset [4] but covers a 10 different genres of both spoken and formal written text. The evaluation in this dataset can be set up to be in-domain (Matched) or cross-domain (Mismatched). We did not include the SNLI data into our training set.

Table 2: Ablation study.

| Method | SQuAD GloVe | | SQuAD ELMo | | IMDB GloVe | MNLI GloVe | |
| | EM | F1 | EM | F1 | Accuracy | matched | mism. |
|---|---|---|---|---|---|---|---|
| GLoMo | **70.84** | **79.90** | **76.00** | **84.13** | **89.16** | **78.32** | 78.00 |
| - decouple | 70.45 | 79.56 | 75.89 | 83.79 | - | - | - |
| - sparse | 70.13 | 79.34 | 75.61 | 83.89 | 88.96 | 78.07 | 77.75 |
| - hierarchical | 69.92 | 79.23 | 75.70 | 83.72 | 88.71 | 77.87 | 77.85 |
| - unit-level | 69.23 | 78.66 | 74.84 | 83.37 | 88.49 | 77.58 | **78.05** |
| - sequence | 69.92 | 79.29 | 75.50 | 83.70 | 88.96 | 78.11 | 77.76 |
| uniform graph | 69.48 | 78.82 | 75.14 | 83.28 | 88.57 | 77.26 | 77.50 |

**Sentiment Analysis** We use the movie review dataset collected in [22], with 25,000 training and 25,000 testing samples crawled from IMDB.

**Transfer Setting** We preprocessed the Wikipedia dump and obtained a corpus of over 700 million tokens after cleaning html tags and removing short paragraphs. We trained the networks $g$ and $f$ on this corpus as discussed in Section 2.1. We used randomly initialized embeddings to train both $g$ and $f$, while the graphs are tested on other embeddings during transfer. We transfer the graph predictor $g$ to a downstream task to extract graphs, which are then used for supervised training, as introduced in Section 2.2. We experimented with applying the transferred graphs to various sets of features, including GloVe embeddings, ELMo embeddings, and the first RNN layer's output.

## 3.2 Main results

On SQuAD, we follow the open-sourced implementation [9] except that we dropped weight averaging to rule out ensembling effects. This model employs a self-attention layer following the bi-attention layer, along with multiple layers of RNNs. On MNLI, we adopt the open-sourced implementation [5]. Additionally, we add a self-attention layer after the bi-inference component to further model context dependency. For IMDB, our baseline utilizes a feedforward network architecture composed of RNNs, linear layers and self-attention. Note the state-of-the-art (SOTA) models on these datasets are [49, 21, 25] respectively. However, these SOTA results often rely on data augmentation [49], semi-supervised learning [25], additional training data (SNLI) [20], or specialized architectures [20]. In this work, we focus on competitive baselines with general architectures that the SOTA models are based on to test the graph transfer performance and exclude independent influence factors.

The main results are reported in Table 1. There are three important messages. First, we have purposely incorporated the self-attention module into **all** of our baselines models—indeed having self-attention in the architecture could potentially induce a supervisedly-trained graph, because of which one may argue that this graph could replace its unsupervised counterpart. However, as is shown in Table 1, augmenting training with unsupervisedly-learned graphs has further improved performance. Second, as we adopt pretrained embeddings in **all** the models, the baselines establish the performance of feature-based transfer. Our results in Table 1 indicate that when combined with feature-based transfer, our graph transfer methods are able to yield further improvement. Third, the learned graphs are generic enough to work with various sets of features, including GloVe embeddings, ELMo embeddings, and RNN output.

## 3.3 Ablation Study

In addition to comparing graph-based transfer against feature-based transfer, we further conducted a series of ablation studies. Here we mainly target at the following components in our framework: decoupling feature and graph networks, sparsity, hierarchical (i.e. multiple layers of) graphs, unit-level objectives, and sequence prediction. Respectively, we experimented with coupling the two networks, removing the ReLU activations, using only a single layer of graphs, using a sentence-level Skip-thought objective [18], and reducing the context length to one [30]. As is shown in Table 2, all these factors contribute to better performance of our method, which justifies our desiderata discussed in Section 2.1.1. Additionally, we did a sanity check by replacing the trained graphs with uniformly sampled affinity matrices (similar to [15]) during the transfer phase. This result shows that the learned graphs have played a valuable role for transfer.

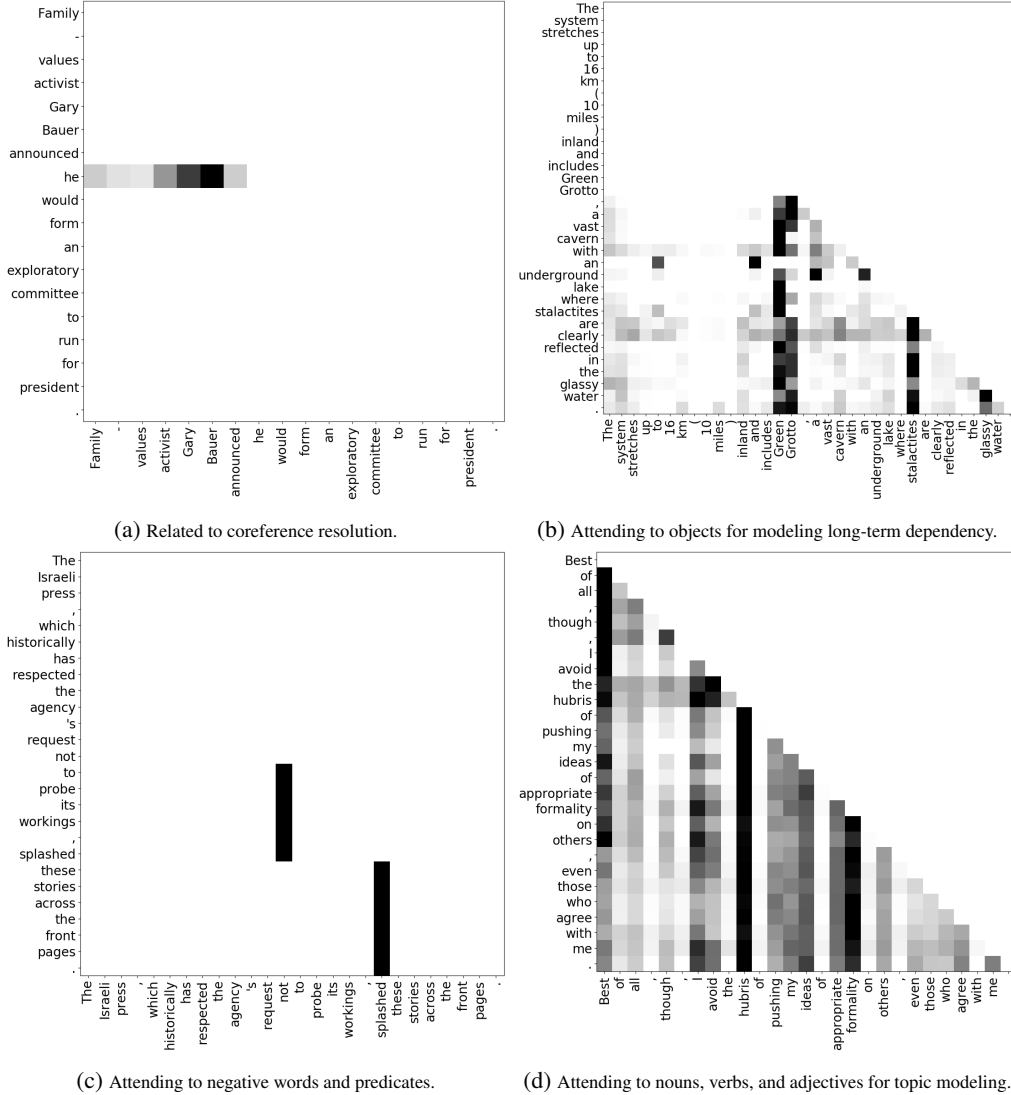

(a) Related to coreference resolution.

(b) Attending to objects for modeling long-term dependency.

(c) Attending to negative words and predicates.

(d) Attending to nouns, verbs, and adjectives for topic modeling.

Figure 3: Visualization of the graphs on the MNLI dataset. The graph predictor has not been trained on MNLI. The words on the y-axis "attend" to the words on the a-axis; i.e., each row sums to 1.

## 3.4 Visualization and Analysis

We visualize the latent graphs on the MNLI dataset in Figure 3. We remove irrelevant rows in the affinity matrices to highlight the key patterns. The graph in Figure 3a resembles coreference resolution as "he" is attending to "Gary Bauer". In Figure 3b, the words attend to the objects such as "Green Grotto", which allows modeling long-term dependency when a clause exists. In Figure 3c, the words following "not" attend to "not" so that they are aware of the negation; similarly, the predicate "splashed" is attended by the following object and adverbial. Figure 3d possibly demonstrates a way of topic modeling by attending to informative words in the sentence. Overall, though seemingly different from human-curated structures such as parse trees, these latent graphs display linguistic meanings to some extent. Also note that the graph predictor has not been trained on MNLI, which suggests the transferability of the latent graphs.

## 3.5 Vision Task

**Image Classification**   We are also prompted to extend the scope of our approach from natural language to vision domain. Drawing from natural language graph predictor $g(\cdot)$ leads the unsupervised training phase in vision domain to a PixelCNN-like setup [27], but with a sequence prediction window

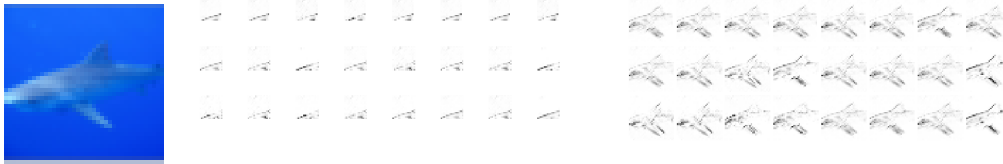

Figure 4: Visualization. Left: a shark image as the input. Middle: weights of the edges connected with the *central* pixel, organized into 24 heads (3 layers with 8 heads each). Right: weights of the edges connected with the *bottom-right* pixel. Note the use of masking.

| Method / Base-model | ResNet-18 | ResNet-34 |
|---|---|---|
| baseline | 90.93±0.33 | 91.42±0.17 |
| GLoMo | **91.55±0.23** | **91.70±0.09** |
| ablation: uniform graph | 91.07±0.24 | - |

Table 3: CIFAR-10 classification results. We adopt a 42,000/8,000 train/validation split—once the best model is selected according to the validation error, we directly forward it to the test set without doing any validation set place-back retraining. We only used horizontal flipping for data augmentation. The results are averaged from 5 rounds of experiments.

of size 3x3 (essentially only predicting the bottom-right quarter under the mask). We leverage the entire ImageNet [11] dataset and have the images resized to 32x32 [27]. In the transfer phase, we chose CIFAR-10 classification as our target task. Similar to the language experiments, we augment **H** by **HM**, and obtain the final input through a gating layer. This result is then fed into a ResNet [13] to perform regular supervised training. Two architectures, i.e. ResNet-18 and ResNet-34, are experimented here. As shown in Table 3, GLoMo improves performance over the baselines, which demonstrates that GLoMo as a general framework also generalizes to images.

In the meantime we display the attention weights we obtain from the graph predictor in Figure 4. We can see that $g$ has established the connections from key-point pixels while exhibiting some variation across different attention heads. There has been similar visualization reported by [50] lately, in which a vanilla transformer model is exploited for generative adversarial training. Putting these results together we want to encourage future research to take further exploration into the relational long-term dependency in image modeling.

## 4   Related Work

There is an overwhelming amount of evidence on the success of transferring pre-trained representations across tasks in deep learning. Notable examples in the language domain include transferring word vectors [29, 24, 30] and sentence representations [18, 10]. Similarly, in the image domain it is standard practice to use features learned in a supervised manner on the ImageNet [33, 37] dataset for other downstream prediction tasks [32]. Our approach is complementary to these approaches – instead of transferring features we transfer graphs of dependency patterns between the inputs – and can be combined with these existing transfer learning methods.

Specialized neural network architectures have been developed for different domains which respect high-level intuitions about the dependencies among the data in those domains. Examples include CNNs for images [19], RNNs for text [14] and Graph Neural Networks for graphs [35]. In the language domain, more involved structures have also been exploited to inform neural network architectures, such as phrase and dependency structures [41, 39], sentiment compositionality [38], and coreference [12, 16]. [17] combines graph neural networks with VAEs to discover latent graph structures in particle interaction systems. There has also been interest lately on Neural Architecture Search [51, 2, 26, 20], where a class of neural networks is searched over to find the optimal one for a particular task.

Recently, the self-attention module [42] has been proposed which, in principle, is capable of learning arbitrary structures in the data since it models pairwise interactions between the inputs. Originally

used for Machine Translation, it has also been successfully applied to sentence understanding [36], image generation [28], summarization [20], and relation extraction [43]. Non local neural networks [44] for images also share a similar idea. Our work is related to these methods, but our goal is to learn a universal structure using an unsupervised objective and then transfer it for use with various supervised tasks. Technically, our approach also differs from previous work as discussed in Section 2.1.1, including separating graphs and features. LISA [40], explored a related idea of using existing linguistic structures, such as dependency trees, to guide the attention learning process of a self attention network.

Another line of work has explored *latent tree learning* for jointly parsing sentences based on a downstream semantic objective [48, 23, 6]. Inspired by linguistic theories of constituent phrase structure [7], these works restrict their latent parses to be binary trees. While these models show improved performance on the downstream semantic tasks, Williams et al [45] showed that the intermediate parses bear little resemblance to any known syntactic or semantic theories from the literature. This suggests that the optimal structure for computational linguistics might be different from those that have been proposed in formal syntactic theories. In this paper we explore the use of unsupervised learning objectives for discovering such structures.

## 5    Conclusions

We present a novel transfer learning scheme based on latent relational graph learning, which is orthogonal to but can be combined with the traditional feature transfer learning framework. Through a variety of experiments in language and vision, this framework is demonstrated to be capable of improving performance and learning generic graphs applicable to various types of features. In the future, we hope to extend the framework to more diverse setups such as knowledge based inference, video modeling, and hierarchical reinforcement learning where rich graph-like structures abound.

## Acknowledgement

This work was supported in part by the Office of Naval Research, DARPA award D17AP00001, Apple, the Google focused award, and the Nvidia NVAIL award. ZY is supported by the Nvidia PhD Fellowship. The authors would also like to thank Sam Bowman for useful discussions.

## Footnotes

[1]Throughout the paper, we use "feature" to refer to unary feature representations, and use "graph" to refer to structural, graphical representations.

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
