[Reviews · NeurIPS 2018]

Reviewer 1



This paper presents a method to transfer graph structures learned on unlabeled data to downstream tasks, which is a conceptual shift from existing research that aims to transfer features (e.g., embeddings). The method consists of jointly training a feature and graph predictor using an unsupervised objective (which are decoupled) and then extracting only the output of the graph predictor for downstream tasks, where it is multiplicatively applied to arbitrary features. The method yields small improvements on a variety of NLP and vision tasks, and the qualitative analysis of the learned graphs does not convince me that it learns "meaningful" substructures. Overall, however, the paper has a compelling and promising idea (graph transfer), and it seems like there is room to improve on its results, so I'm a weak accept. Detailed comments: - Is "unsupervisedly" a word? It sounds weird... - The objective function in eq 3 is interesting and could have potential uses outside of just graph induction, as it seems especially powerful from the ablations in table 2... Even for feature learning (e.g., ELMo), predicting the next D words instead of just the next word may give gains. However, I'd like more details on how exactly the RNN decoder was implemented. Did the authors feed the ground-truth previous word/pixel at every step (making the decoder a language model) or try and predict the next D words/pixels directly from the hidden state at time t (e.g., no input feeding)? The latter method seems like it would result in more powerful representations, as the former gives the decoder a lot of leeway to ignore parts of the original context after a couple predictions. - Did you try experimenting with the features learned by your feature predictor as embeddings (instead of GloVe or ELMo)? Perhaps those features are even better because they were jointly trained with the graph predictor? - The decoupling between feature and graph predictor is not very clearly described. It appears like they are not entirely decoupled, as the feature predictor takes the learned graph as input, and the graph predictor is only trained with one type of feature predictor... Fig 2 is also confusing, as there are arrows from the graph predictor to the feature predictor but the objective is computed over the "mixed graph". It would be easier to understand if the graph and feature predictor diagrams were switched, and the "mixed graph" explicitly shown as a combination of graph * weighted feature sum at each layer. - Instead of performing transfer with just a single graph M as in sec 2.2, it seems like you could also try using each layer-specific affinity matrix individually (e.g., if your downstream model also had multiple layers). - Fig 3 is very confusing. Are these visualizations of M (the mixture of graphs)? How does each row sum to 1 if many of them are completely blank? A color scale would be very helpful here, and also a more informative caption.

Reviewer 2



The work proposes a noval nn structure to unsupervisedly learn the underlying graph structures that describe the correlations between features. The extract graph structures is transferable and may help with multiple downstream supervised learning tasks. I am not familiar with the related works while the authors describe clearly the relation to the previous works. They also give solid reasons on the proposed tricks that are used to build up the new network and justify the effectiveness of them via extensive experiments. Hence, the contribution reads non-trivial. However, I have some doubt that prohibits me from giving strong recommendation. First, I think the method only works when the embedding methods of unsupervised part and supervised part are same/at least similar to each other, while the work claims that the embedding approach of the supervised part can be arbitrary. Apparently, the learnt graph strongly depends on the embedding layer of the unsupervised part due to the way that it is trained. As for this concern, I would like the authors to clarify the results obtained in Table 1. As there are different embedding methods used in the supervised part, do they share the same graph learnt from the unsupervised section? Second, essentially, the learnt graph performs as the correlation between features. So I think it should be more fair to add some cross layers within the baselines to make comparison, where the cross layers are learnt only within the supervised section. Only in this way, the effectiveness of transferability can be well established.

Reviewer 3



This paper argues for pre-training not just embeddings but also a task-independent graph structure, inspired by (but different to) self-attention. It's a neat idea and I believe we need to pre-train more of our models in NLP so it's the right direction for research. Paper is well written and the technical description is clear. Some nit-picky comments: After reading the abstract and intro I am still a little confused about what the paper is trying to do. Work from last year started to pre-train/learn more than single word vectors. In particular, at last year's NIPS, CoVe (Context Vector by McCann et al) was introduced which pre-trained an LSTM with translation. This was then later extended with ELMO at EMNLP this year. 2.1.1 Desiderata This should probably be a \subsection instead